# Strontium-Modified Scaffolds Based on Mesoporous Bioactive Glasses/Polyvinyl Alcohol Composites for Bone Regeneration

**DOI:** 10.3390/ma13235526

**Published:** 2020-12-03

**Authors:** Javier Jiménez-Holguín, Alvaro López-Hidalgo, Sandra Sánchez-Salcedo, Juan Peña, María Vallet-Regí, Antonio J. Salinas

**Affiliations:** 1Department of Química en Ciencias Farmacéuticas, Faculty of Farmacia, Instituto de Investigación Hospital 12 de Octubre, imas12, Universidad Complutense, UCM, 28040 Madrid, Spain; javiej03@ucm.es (J.J.-H.); alvlop06@ucm.es (A.L.-H.); juanpena@ima.ucm.es (J.P.); vallet@ucm.es (M.V.-R.); 2Networking Research Center on Bioengineering, Biomaterials and Nanomedicine, CIBER-BBN, 28040 Madrid, Spain

**Keywords:** mesoporous glass scaffolds, SrO, cytocompatibility, pre-osteoblastic cells

## Abstract

In the search of a new biomaterial for the treatment of bone defects resulting from traumatic events, an osteoporosis scenario with bone fractures, tumor removal, congenital pathologies or implant revisions for infection, we developed 3D scaffolds based on mesoporous bioactive glasses (MBGs) (85 − x)SiO_2_–5P_2_O_5_–10CaO–xSrO (x = 0, 2.5 and 5 mol.%). The scaffolds with meso-macroporosity were fabricated by pouring a suspension of MBG powders in polyvinyl alcohol (PVA) into a negative template of polylactic acid (PLA), followed by removal of the template by extraction at low temperature. SrO-containing MBGs exhibited excellent properties for bone substitution including ordered mesoporous structure, high textural properties, quick in vitro bioactive response in simulated body fluid (SBF) and the ability of releasing concentrations of strontium ions able to stimulate expression of early markers of osteoblastic differentiation. Moreover, the direct contact of MC3T3-E1 pre-osteoblastic cells with the scaffolds confirmed the cytocompatibility of the three compositions investigated. Nevertheless, the scaffold containing 2.5% of SrO induced the best cellular proliferation showing the potential of this scaffold as a candidate to be further investigated in vitro and in vivo, aiming to be clinically used for bone regeneration applications in non-load bearing sites.

## 1. Introduction

In the case of serious injury, bone tissue is not able to regenerate such damage, and a bone graft is required. The gold standard is bone autograft since it gathers all properties needed in all bone-grafting materials such as osteoinduction, osteogenesis and osteoconduction [1,2]. The problems related to autografts are the limited availability of tissues for this purpose and that the harvest of these materials are associated with severe pain in the operated region and are prone to bacterial adhesion as a result of surgery and prosthetic rejection [3,4].

Due to the economic cost and poor availability of tissues for bone grafting, several artificial biomaterials have been developed to replace autografts, many of them being bioceramics [5]. In this family of compounds, mesoporous bioactive glasses (MBG) of the SiO_2_–CaO–P_2_O_5_ system represent a group widely proposed as optimum candidates for bone substitution because the inductive effect of Si(IV), P(V) and Ca^2+^ ions released to the surrounding medium [5,6]. MBGs exhibit high surface area and pore volume that enhance the in vitro bioactive response [7,8] and enable promoting the proliferation and differentiation of osteogenic related cells [9], as well as hosting different molecules with biological activity [10,11]. Furthermore, silanol groups (Si−OH) on the surface of these materials allow the functionalization with diverse groups like amines and carboxylic acids, among others, to modulate their response in the biological medium [12,13]. An additional property of these glasses is that they allow the incorporation in their structure of therapeutic ions such as copper, gallium, cerium, zinc, strontium etc. [14,15,16,17,18,19,20,21,22], to be progressively released, acting as ion control-delivery systems. Although strontium is not considered to be an essential element, it has been demonstrated that strontium also promotes both osteogenic proliferation and differentiation through several pathways such as Wnt/β-Catening signaling [23], Ras/MAPK signaling [24] and the Ca sensing receptor (CasR) [25]. On the other hand, it is well known that strontium has the ability to inhibit osteoclastogenesis, blocking the interaction between Rank ligand and its receptor [26]; Strontium also inhibits osteoclasts differentiation [27] and promotes osteoclasts apoptosis mediated by CasR [28].

A recent approach for the release of therapeutic ions and molecules is the use of three-dimensional scaffolds obtained by 3D printing [29]. For this purpose, a paste based on MBG and a binding agent such as polyvinyl alcohol (PVA) is commonly used due to its variety of applications within the field of tissue engineering [30]. However, in many cases the preparation of a suitable paste for extrusion thereof is difficult and laborious to optimize.

Taking these points into account, the application of this system as a three-dimensional scaffold has been proposed using a slurry based on MBG/PVA and a negative mould obtained by 3D printing [31,32], employing polylactic acid (PLA) that was removed by extraction to obtain a macroporous scaffold.

Several biomaterials were developed for the treatment of bone defects resulting from traumatic events, an osteoporosis scenario with bone fractures, tumor removal, congenital pathologies, and implant revisions of infection. With this aim, this paper describes the synthesis, characterization and in vitro cytocompatibility in pre-osteoblasts cultures of the SrO-containing meso-macroporous MBG scaffolds. With this purpose, three MBGs with composition (in mol.%) (85 − x)SiO_2_–5P_2_O_5_–10CaO–xSrO (x = 0, 2.5 and 5) were investigated. After the microstructural characterization, MBG powders were compacted into disks to evaluate, in vitro, their bioactive response and ions release capability. Finally, the cytocompatibility of MBG/PVA 3D scaffolds was assessed in MC3T3-E1 pre-osteoblastic cell cultures.

Therefore, the final goal of this study was twofold, on the one hand to check whether this less common processing method would allow us to obtain useful meso-macroporous scaffolds for bone tissue engineering applications in non-load bearing sites and, above all, to check whether the release of strontium ions contained in the scaffolds would produce advantages for this clinical application.

## 2. Materials and Methods

### 2.1. Synthesis of the Mesoporous Bioactive Glasses

Three MBGs (85 − x)SiO_2_–10CaO–5P_2_O_5_–x SrO, (x = 0, 2.5 and 5 mol.%) were synthesized by the evaporation-induced self assembly (EISA) method. HNO_3_ 0.5 M was used as a catalyst and Ca(NO_3_)_2_∙4H_2_O, tetraethyl orthosilicate (TEOS), triethyl phosphate (TEP) and Sr(NO_3_)_2_ as calcium, silica, phosphate and strontium sources, respectively. (The reactants were purchased from Sigma-Aldrich, St. Louis, MO, USA).

Briefly, the synthesis was performed following the procedure previously published [33,34], by mixing Pluronic^®^ F127, ethanol (99.98%), distilled water and HNO_3_ in a glass flask covered with Parafilm^®^ and kept 1 h under stirring. The amounts of each reactant (Table 1) were added sequentially at 60 min intervals; the mixture was kept 14 h at 40 °C under stirring. Afterwards, in order to evaporate the ethanol, the mixture was poured into open Petri dishes and kept 4 d at 30 °C in a stove to evaporate the solvent. Throughout this process, the critical micellar concentration was reached, allowing the mesoporous phase to be obtained. The MBG powders were obtained after heating the so obtained laminar dry gels for 6 h at 700 °C (heating ramp of 1 °C/min) and crushing and sieving with a 40 μm mesh, giving rise to 0, 2.5 and 5Sr-MBG powders.

### 2.2. Characterization of MBGs

To evaluate the physicochemical characteristics of the materials and verify their suitability for the objective of our study, MBG powders were initially characterized by small-angle X-ray diffraction, SA-XRD, in a X’pert-MPD system (Eindhoven, The Netherlands) equipped with Cu Kα radiation in the 0.6° to 8° 2θ range; by thermogravimetric and differential thermal analysis (TG/DTA) from 30 °C to 900 °C with a ramp of 5 °C/min (air flow: 100 mL/min) in a Perkin Elmer Pyris Diamond system (Waltham, MA, USA); by Fourier transformed infrared spectroscopy (FTIR) in a Thermo Scientific Nicolet iS10 apparatus (Waltham, MA, USA) equipped with a SMART Golden Gate attenuated total reflection (ATR) diffuse reflectance accessory and by scanning electron microscopy (SEM) in a JSM-7600F (JEOL) microscope (Tokyo, Japan), which operates between 5 and 15 kV, coupled with an energy dispersive X-ray spectroscopy (EDS) system (Oxford Instruments, Abingdom, UK). Moreover, MBG powders were analyzed by transmission electron microscopy (TEM) in a JEM-2100 JEOL microscope operating at 200 kV (Tokyo, Japan). Furthermore, the composition of the sample was determined by energy dispersive X-ray spectrometry (EDS) with an Oxford LINK EDS analyzer coupled to the TEM microscope.

On the other hand, the MBGs texture was characterized by nitrogen adsorption using a Micromeritics 3 Flex (Norcross, GA, USA) that allowed obtaining the surface area, SBET, by the Brunauer–Emmett–Teller (BET) method [35], and pore size distributions by the Barret–Joyner–Halenda (BJH) method [36]. Finally, ^29^Si and ^31^P solid state single pulse magic angle spinning nuclear magnetic resonance (SP MAS-NMR) were performed in a Bruker Avance AV-400WB spectrometer (Karlsruhe, Germany) equipped with a 4 mm zirconia rotor. Frequencies used were 79.49 and 161.97 MHz for ^29^Si and ^31^P, respectively, and chemical shift values were referenced to tetramethylsilane for ^29^Si, and H_3_PO_4_, for ^31^P. Nuclear magnetic resonance (NMR) spectra were assembled with the OriginLab 7.0 software (OriginLab Corporation, MA, USA). The quality of the adjustment of the spectra was verified by comparing the agreement of the experimental and calculated spectra.

### 2.3. In vitro Studies in Acellular Solutions

In vitro studies were performed on disks (10 mm diameter, 1 mm high) obtained by compacting 100 mg of powdered MBG with 5 MPa of uniaxial pressure for 1 min. Surface reactivity was assessed by monitoring the formation of an apatite-like layer on the surface of the MBG disks after immersion in simulated body fluid (SBF) [37]. In addition, the release of ions was investigated by soaking the disks in Eagle’s minimal essential medium (MEM).

The MBG disks were previously sterilized for 20 min under UV radiation (10 min per side) in a laminar flow cabinet and the SBF solution was filtered through a 0.22 µm filter to avoid bacterial contamination. MBG disks were immersed for 8, 24 and 72 h in 30 mL of SBF at 37 °C, pH 7.4. The volume of SBF was chosen according to previous studies [38], to keep the ratio V_S_ = D_S_/0.075 constant, where V_S_ is the volume of SBF (mL) and D_S_ is the external geometric area (cm^2^) of the sample. MBG disks were immersed, hanging from the lid with a platinum wire, in the SBF medium within polypropylene flasks [16]. Tests were carried out with two replicates per material and a control of SBF without material. After the test finished, disks were removed and rinsed with acetone and ethanol (to stop the apatite formation reaction and to dry the disks). The disks were stored in an oven at 60 °C for further characterization [39]. The formation of an apatite-like phase was characterized by FTIR spectroscopy and SEM.

Ion release assay from MBG disks was performed by placing each disk on a plate with transwell immersed in 2 mL of MEM with 0.5% of streptomycin/penicillin (Sigma-Aldrich. St. Louis, MO, USA). This allowed obtaining aliquots of medium without MBG grains. An aliquot was extracted after 8 h and each day for one week. The cumulative concentrations of Ca(II), P(V) and Sr(II) in the culture medium were obtained by Inductively Coupled Plasma/Optical Spectrometry (ICP/OES) using an OPTIMA 3300 DV device (Perkin Elmer, Waltham, MA, USA). Three measurements were made on each of the two replicates per composition examined.

### 2.4. Processing of MBG Scaffolds

To fabricate the scaffolds, a 15% of PVA (Sigma Aldrich, St. Louis, MO, USA) suspension was prepared by mixing 1.25 g of PVA in 10 mL of miliQ H_2_O at 90 °C under magnetic stirring [40]. The addition of the polymeric additive (PVA) was necessary to obtain a workable paste [30,41]. Then, 5 mL of the PVA solution was introduced in an Intertronics Think ARE-250 automatic mixer (Oxfordshire, England), together with six agate mixing balls (1 cm in diameter). After that, MBG component was sequentially added in four mixing cycles (1.38, 1.38, 0.69 and 0.69 g) to ensure an adequate viscosity and homogeneity of the paste. On the other hand, the negative templates and external moulds were fabricated with PLA by using a 3D printer (Sigma BCN3D printer, Barcelona, Spain). The PVA/MBG paste was introduced into the PLA negative template with a syringe that facilitated the penetration of such paste within the voids of the PLA negative template. This ensemble was left to dry for 4 h, removed from the external mould and lyophilized in a Telstar LyoQuest freeze dryer (Tarrasa, Spain). This technique ensured the complete drying of the resulting scaffold without a noticeable shrinkage. The final step of this procedure involved the elimination of the PLA negative mould by successive washes in dichloromethane (Sigma Aldrich, St. Louis, MO, USA). The complete elimination of such solvent was carried out in a vacuum oven at 40 °C for 48 h (Scheme 1).

### 2.5. In vitro Cytocompatibility Assays

All these assays were performed on the PVA/MBG scaffolds irradiated with UV light for 5 h on each side. Pre-osteoblastic mouse cells MC3T3-E1 (subclone 4, CRL-2593; ETCC, Mannassas, VA, USA) were cultured in supplemented medium, α-MEM with 10% fetal bovine serum (FBS, Gibco, Thermo Fisher Scientific, Wilmington, DE, USA), 1% penicillin-streptomycin and 5 mM of L-glutamine (Gibco, Thermo Fisher Scientific, Wilmington, DE, USA), in a 5% CO_2_ atmosphere at 37 °C. Cells were washed with phosphate-buffered saline (PBS, pH 7.4 (Gibco, Thermo Fisher Scientific, Wilmington, DE, USA), 1% penicillin-streptomycin and 5 mM L-glutamine and collected using 5 mL of 0.25% trypsin-EDTA (Gibco, Thermo Fisher Scientific, Wilmington, DE, USA), 1% penicillin-streptomycin and 5 mM L-glutamine. Cells suspensions were centrifuged 10 min at 1200 rpm. The resulting material was resuspended with fresh medium and 2.5·10^5^ cells were seeded onto the scaffold top surface for the analysis of pre-osteoblastic cells viability, proliferation, cytotoxicity and expression of markers of osteoblastic differentiation. Three controls and three replicates per sample were carried out and incubated at 37 °C in a 5% CO_2_ atmosphere.

#### 2.5.1. Morphological Studies by Confocal Laser Scanning Microscopy

Cell morphology was studied using fluorescence microscopy with a confocal scanning microscope OLYMPUS FV1200 (OLYMPUS, Tokyo, Japan) using a 60X FLUOR water dipping lens (NA = 1) after 24 h. Cells were incubated 24 h at 37 °C in a 5% CO_2_ atmosphere. After incubation, media were removed and washed twice with PBS. Then, cells were fixed with 75% ethanol and stained with Phalloidin-ATTO 565 (dilution 1:40, Molecular Probes) for 10 min. Next, enough volume to cover the scaffolds of Fluoroshield^®^ with DAPI (1:1000, Sigma Aldrich, St. Louis, MO, USA) was added for 10 min. Thereafter, each plate was washed twice and remained with PBS. DAPI and Atto 565-phalloidin staining were shown in blue and red, respectively. Micrographs were obtained using FV10-ASW (4.2 Version, Waltham, MA, USA) to construct a single 2D image, which was converted into a TIF image file using multiple images obtained from each section in the Z axis using an algorithm that shows the maximum value of pixels of each section in Z for each 1 μm.

#### 2.5.2. Proliferation Assay and Cytocompatibility Assay

The pre-osteoblastic cell growth onto the scaffolds was determined by fluorescence intensity with a Synergy4 Multimode Plate Reader (BioTek Instruments, Winooski, VT, USA) at 1, 3, 7 and 10 d with excitation and emission wavelengths of 560 and 590 nm, respectively. Alamar Blue method (AbD Serotec, Oxford, UK) was used, by following the manufacturer’s instructions. Scaffolds were exposed to 1:10 AlamarBlue^®^ (Invitrogen, Thermo Fisher Scientific, Wilmington, DE, USA) solution for 4 h in darkness before fluorescence measurements.

The medium used for culture cells at 7 d was used for the lactate dehydrogenase (LDH) test (Spinreact, Girona, Spain). The mixture of medium and reagent was measured after 7 d of cell seeding at 340 nm every 60 s for 4 min in a Synergy4 Multimode Plate Reader.

#### 2.5.3. Real Time PCR

Total RNA was isolated from cells seeded onto the scaffolds after 10 d of cell culture (Trizol, Invitrogen, Groningen, The Netherlands). Gene expression of two osteoblastic differentiation markers, alkaline phosphatase (ALP) and runt related transcription factor-2 (RUNX2) were analyzed by real time PCR using QuantStudio5 equipment and a described protocol (Applied Biosystems-Thermo Scientific, Foster City, CA, USA) [39]. TaqMan^MGB^ probes were obtained by Assay-by-Design^SM^ (Applied Biosystems). For each sample and using the cycle threshold (Ct) value, mRNA copy numbers were calculated and glyceraldehyde-3-phosphate dehydrogenase (GAPDH) rRNA (a housekeeping gene) was amplified in parallel with the tested genes. The number of amplification steps required to reach an arbitrary Ct was computed. The relative gene expression was represented by 2^−ΔΔCt^, where ΔΔCt = ΔCt_target gene_ − ΔCt_GAPDH_. The fold change for the treatment was defined as the relative expression compared with the control GAPDH expression, calculated as 2^−ΔΔCt^, where ΔΔCt = ΔC_treatment_ − ΔC_control_.

### 2.6. Statistical Analysis

Results were expressed as the mean ± SEM (standard error of the mean) of 6 specimens split in two independent experiments (3 separate groups of samples in each one). Statistical analysis was performed with the nonparametric Kruskal–Wallis test and post-hoc Dunn´s test. A value of *p* < 0.05 was considered significant.

## 3. Results and Discussion

### 3.1. Microstructural Characterization of MBGs

First, EDS and FTIR studies allowed confirming the complete elimination of the surfactant Pluronic^®^ F127 used during the MBG powders synthesis, a critical determination before using these materials for the in vitro studies. Moreover, the powders that will be termed along the study as 0, 2.5 and 5Sr-MBG were characterized by TEM and SA-XRD to confirm the preservation of the mesoporous structure.

As shown in Figure 1, all samples exhibited high mesopore order, even those doped with the highest SrO content (5%). As it is observed, the three samples exhibited mesoporous order, including 5Sr-MBG where the image was taken in the perpendicular orientation, to check that in this case the order was also preserved. Moreover, the strontium content proportionally increased from 2.5Sr-MBG to 5Sr-MBG. as shown in the inset spectra of Figure 1 and in Table 2 that includes the atomic percentages of each element obtained by EDS. As it is observed, the nominal atomic compositions used in the synthesis process were close to those obtained experimentally, except for 5Sr-MBG that has an atomic composition that is 22% less than the nominal one, i.e., the SrO content was 3.7% instead of the intended 5%.

Well-defined XRD spectra in Figure 2A confirm the mesoporous order [37]. However, as can be observed, the mesoporous order decreased with the strontium additions, being the least ordered with the highest SrO content, i.e., 5Sr-MBG sample. Table 2 shows the interplanar distances (d_hk_) for the direction (10). The unit cell (a_0_) was then calculated by the formula a_o_ = 2d_10_/√3. The term a_0_ is the distance between pore centers and can be obtained by crystallographic determinations according to a hexagonal unit cell of the P6m planar group [42,43]. Moreover, the wall thickness of the mesoporess (t_wall_) can be calculated by comparing a_o_ with the pore size value (D_P_) obtained by nitrogen adsorption analysis. For hexagonal p6mm structures t_wall_ = a_o_ − D_P_.

Figure 2B shows the N_2_ adsorption isotherms for the MBG samples. 0Sr-MBG exhibits the characteristic type IV adsorption isotherm of a mesoporous material that varies due to the addition of Sr to the structure. The hysteresis cycle of Sr-free MBG shows the H1 type cycle, associated with porous materials with well-defined cylindrical pore channels. For strontium doped MBGs H2 type hysteresis cycles were obtained, indicative of porous materials that become disordered when the distribution and size of the pores is not well defined, and it is also distinctive of materials with pores in the form of an inkwell [44]. Considering pore diameter distribution (Figure 2C), the presence of mesopores was observed for all samples, but micropores were only detected for the Sr-free sample.

Table 2 also includes the textural properties of the powders, BET surface (S_BET_), pore volume (V_P_) and pore diameter (D_P_). As is observed, the increase of Sr in the MBGs structure decreased and consequently increased t_wall_, which finally induced a noticeable decrease on S_BET_ (from 307 to 141 m^2^/g), V_p_ (from 0.42 to 0.19 cm^3^/g) and D_P_ (from 7 and 5.7 nm). These results confirmed previously reported results indicating that the incorporation of Sr^2+^ ions in MBG causes a partial loss of textural properties [15,16,19,21,45]. This loss can be explained by the incorporation of strontium ions to the silica network as network modifiers, replacing calcium ions. Strontium ions are larger than calcium ones, making the silica lattice thicker and less dense, as can be deduced from Table 2. This could be the same reason why we found a decrease in the mesoporous order (Figure 2A).

To study the atomic environment of the silica matrix of samples, ^29^Si and ^31^P NMR measurements were performed. Figure 3 collects the ^29^Si and ^31^P solid state NMR spectra of the different samples; Q^2^, Q^3^ and Q^4^ represent the silicon atoms (indicated by Si*) in (NBO)_2_Si*−(OSi)_2_, (NBO)Si*−(OSi)_3_, and Si*(OSi)_4_ (NBO = non-binding oxygen), respectively. The signals at −111 to −112 ppm were assigned to Q^4^, at −103 to −104 to Q^3^ and at −95 to −98 to Q^2^ [46].

Table 3 includes the chemical shifts (CS), relative areas of the peaks corresponding to ^29^Si and ^31^P, respectively, and the silica network connectivity, <Q^n^>. The NMR spectra in Figure 3 reveal that when the SrO content was increased, a huge Q^4^ percentage decreased together with a parallel increase of Q^3^. Meanwhile, the network connectivity <Q^n^> of samples decreased from 3.826 to 3.474. These results indicate that strontium incorporates to the MBG structure and acts as a network modifier since it inhibits silanol condensation, thus favoring network depolymerisation according to the proposed model for the modified silica network structure [47], as previously mentioned.

Concerning ^31^P solid state NMR spectra, Q^0^ and Q^1^ represent, respectively, the phosphorus atoms (indicated by P*) in (NBO)_3_P*−(OP) and (NBO)_2_−P*−(OP)_2_. The signals at 1.8 to 2.2 ppm were assigned to Q^0^, the typical environment of amorphous orthophosphate, while signals around −7 ppm were attributed to Q^1^ tetrahedra [46]. Q^0^ values increased up to 96.5 and, consequently, Q^1^ decreased from 6.5 to 3.4 as a consequence of the strontium ions increment in the silica matrix that led to a decrease of P–O–Si units.

This characterization confirmed that strontium inclusion causes a less polymerized structure as can be deduced from the detected decrease of the silicon–oxygen and phosphorous–oxygen interpenetrating networks. Accordingly, bioactive glasses became more amorphous and, thus, they can be tailored as a function of strontium additions [38].

### 3.2. In Vitro Studies in Acellular Solutions

The bioactivity tests of the MBG disks consisted of monitoring the deposition of a nanometric carbonate hydroxyapatite (n-CHA) layer after being immersed 8, 24 and 72 h in SBF by FTIR and SEM. This process begins with the formation of silanol groups on the MBG surface due to the ionic interchange between the ions from the glass and protons in the SBF. Next, the silanol groups begin to condense and the surrounding environment becomes acidic. This decrease in pH guarantees nucleation so that amorphous calcium phosphate (a-CaP) begins to precipitate, and over time it matures to n-CHA [48].

FTIR spectra of MBGs before soaking displayed in Figure 4 are characterized by two strong absorption bands: one at 400–500 cm^−1^ and another one at 1100–1000 cm^−1^, corresponding to the asymmetric bending vibration of the Si−O−Si bond and a third strong band at around 800 cm^−1^ of the symmetrical stretching vibration of the Si−O bond [48].

The formation of the n-CHA layer can be detected by the presence of two bands at 600 and 570 cm^−1^, indicative of the presence of the P−O vibration in a crystalline environment, and another two bands at 1420 and 1450 cm^−1^, attributable to carbonate ions (CO_3_^2−^). In the case of 0Sr-MBG, both types of evidence can be clearly observed within the first 8 h of immersion. The Sr inclusion retards the mineralization process as can be deduced from the delayed observation of these bands: the double peak of crystalline P−O cannot be detected until 24 h for the 2.5Sr-MBG sample and 72 h for the 5Sr-MBG sample. However, the first evidence of carbonate is observed only after one day in both samples.

The morphological modifications caused by the formation of the n-CHA layer can be observed in Figure 5, which collects the SEM micrographs as well as the EDS spectra at the different soaking times. As expected from the FTIR results, 0Sr.MBG shows at 8 h the presence of a newly formed layer composed by nanometric needle-like crystals of n-CHA. When the soaking time increases, the thickness of the coating increases, as can be deduced from the progressive diminution of the Si signal, compared to the Ca and P ones, in the EDS spectra. SEM and EDS analysis also confirmed the FTIR results, showing a delay in the formation of needle-like crystals layer for 2.5Sr-MBG and 5Sr-MBG samples that were detected at 24 and 72 h, respectively. This fact probably occurs due to the replacement of the Ca^2+^ ions in the silica network by Sr^2+^ ions, which do not form apatite nucleation canters, slowing down the formation of the n-CHA layer in the surface of the samples.

Therefore, we can conclude that strontium retards but does not inhibit the formation of the n-CHA layer in SBF, the in vitro acellular test usually employed to characterize a bioactive response.

Since the ions concentration around the implanted material is critical for its adequate integration and to exert desired biological actions, the cumulative release of calcium, phosphorous and strontium ions after soaking the MBGs for 1, 3 and 7 d in MEM was studied (Figure 6). The three samples released similar amounts of Ca^2+^ ion that increased with the amount of strontium. This fact can be attributed to their more amorphous structure observed by NMR (Figure 3) and, consequently, higher reactivity, i.e., higher solubility. A similar tendency should be expected for the leaching of the phosphorous ions. However, the low values detected for these ions seem to contradict this increasing reactivity. The formation of strontium silicates or phosphate salts in 5Sr-MBG, as reported by Kaur et al. [49], could explain the apparent contradiction between the high values of Ca and the low values of P despite the more “open” structure, as a result of the strontium presence in the samples. Finally, 5Sr-MBG samples released Sr^2+^ ions two-fold higher than 2.5Sr-MBG samples, as expected, up to 156 mg/L; sufficient quantities to exert biological activity but within the values that do not cause toxicity, since these quantities can be metabolized and expelled through the urine [50].

### 3.3. MBG Scaffolds Characterization

The design and fabrication of meso-macroporous Sr-MBG/PVA scaffolds, with a 3D printed PLA negative template and an external mould, using a low temperature extraction method, were carried out with a modified method previously reported [40,51]. Briefly, 3D printing is a computer-controlled fabrication technique that uses fused filament fabrication for generating 3D objects by shaping successive cross-sectional laminae, one on top of other parts by deposition of molten PLA material. The optimized PVA/MBG paste was introduced into the PLA structure and freeze-dried to maintain their macroporosity inside PVA/MBG composite; finally, PLA negative template was removed by solution in dichloromethane avoiding any calcination and sintering process. This process leads to obtaining meso-macroporous scaffolds 0, 2.5 and 5Sr-MBG with a weight ratio of 77 ± 2/23 ± 2% of MBG/PVA respectively.

SEM micrographs of the MBG scaffolds with their respective EDS spectra are shown in Figure 7; a homogeneous integration of the MBG particles within the PVA polymeric matrix, both at the surface and within the bulk, was observed in the three MBG scaffolds. The sizes of the channels formed by the PLA negative template elimination are determined to be around 0.5 mm in diameter. Besides, an intergranular macroporosity (<10 µm) obtained by water removal by freeze-drying can be observed. This bimodal pore distribution (macropores and giant channels) is suitable for a rapid vascularization and osteoconduction essential for bone remodeling. The EDS spectra depicted below confirmed the homogeneous incorporation of strontium, deduced from the 10 analyses effectuated on each sample, confirming an increasing Sr doping into the MBG structure (Table 2).

### 3.4. Cytocompatibility Assays on MBG Scaffolds

Once the samples were characterized from the microstructural point of view, cytocompatibility tests were performed to study the capacity of these MBG materials to stimulate MC3T3-E1 proliferation and the expression of early markers of osteoblastic differentiation onto MBG scaffolds. This pre-osteoblastic cell line was, as well, used to evaluate the scaffolds toxicity.

3D confocal microscopy tests were carried out on the 2.5Sr-MBG scaffolds after 24 h of MC3T3-E1 seeding (Figure 8A). The geometrical center of each cylindrical scaffold was scanned to determine the cell morphology. Atto 565-phalloidin was used as the fluorescence probe of F-actin microfilaments and DAPI fluorostaining for nuclei was used. Actin is the most important interconnected filament protein in the cytoskeleton, performing critical cell functions. Representative images of preosteoblasts attached onto 2.5Sr-MBG scaffolds, showing their typical spindle-shaped morphology [52]; they were adequately spread on the scaffold surface, indicating their cytocompatibility as observed in the figure.

The characterization of the influence of strontium on the cytocompatible behavior of the MBG scaffolds as well as on the proliferation and gene expression of early markers of osteoblastic differentiation is depicted in Figure 8 and Figure 9. The proliferation assay performed for the MBG scaffolds shows a noticeable increment from 7 days in the 2.5Sr-MBG scaffolds, even exceeding the growth observed for cell control (Figure 8B). Regarding the LDH measurement (Figure 8C), the values obtained are not significantly different when compared to the control. Despite there being minor differences, the slightly higher values observed for 2.5Sr-MBG scaffolds were no significant. Therefore, we can conclude that none of the MBGs investigated were cytotoxic.

Figure 9A,B collect the gene expression characterization results of two markers involved in osteogenic of differentiation, measured by real-time PCR: ALP and RUNX2. 2.5Sr-MBG scaffolds did significantly increase the ALP gene expression markers; meanwhile, it can be observed that there is a slight decrease in the expression of the ALP gene for the 5Sr-MBG sample, but without being significant. Considering the results of ion release (Figure 6), it can be seen that the 2.5Sr-MBG disk sample releases a concentration close to 70 mg/mL after 7 d, while the 5Sr-MBG disk sample release is more than double that concentration. Although these samples, when forming scaffolds, will release a slightly lower amount than that obtained with the disks, we can approximate the optimal concentration range for proliferation, as we can deduce it from scaffold proliferation tests. Taking into account this point and the optimal therapeutic concentrations studied in the literature of around 30 mg/L [22], we can deduce that the optimal concentration is between 30 and 60 mg/L for this cell line. This may be the reason why cells have proliferated with the 2.5Sr-MBG scaffold sample almost twice as much as with the 5Sr-MBG scaffold sample, and we see the same difference in ALP gene expression for the same samples.

On the other hand, none of the materials studied significantly alters the gene expression of RUNX2, an early marker of osteoblastic differentiation (*p* < 0.05).

In 2010, Gentleman et al. [53] reported the putative benefits of strontium-substituted bioglasses. These materials stimulated an increase in the osteoblasts metabolism and inhibited osteoclast differentiation and resorption via the release of these ions, but also promoted osteoblast proliferation and ALP activity through a direct contact with the cells. For other bioceramics, such as calcium phosphate silicates, [54] it was observed that ALP osteogenic differentiation of rBMSCs increased after 7 d. An increase was also observed for RUNX2, after 4 d in strontium-substituted calcium phosphate silicate compared with calcium phosphate silicate, although with higher amounts of strontium than in our Sr-enriched MBGs. Moreover, Liu et al. [55] demonstrated that the expression levels of the osteogenic-related genes of bone marrow stromal cells (BMSCs) for the 3D printed poly(ε-caprolactone)/SrHA scaffolds were up-regulated compared to that of 3D printed poly(ε-caprolactone)/HA scaffold up to 14 d. Finally, Zhang et al. [22] showed that the osteogenic-related gene expression—RUNX2, osteocalcin (OCN) and bone sialoprotein (BSP) mRNA expression—of MC3T3-E1 cells onto Sr-substituted MBG scaffolds promoted osteogenic differentiation of MC3T3-E1 cells after 7 and 14 d. Recently, Kaur et al. [49] synthesized sol-gel SiO_2_–CaO–MgO–P_2_O_5_ glasses doped with up to 5 mol.% SrO with no significant differences of proliferation, attachment and metabolic activity of osteosarcoma cells. This was attributed to the higher surface in Sr-containing samples that was fourfold lower than the MBGs investigated in the present paper. Compared with these materials, our Sr-doped MBGs exhibited a higher proliferation and early markers of osteoblastic differentiation with lower strontium content. This effect could be attributed to a higher and faster release of Sr ions from the MBGs described in this work (75 mg/mL in 2.5Sr-MBG and 160 mg/mL in 5Sr-MBG) compared to others that reached ca. of 35 mg/mL for 20% of Sr-MBG substitution and 85 mg/mL for maximum SrO-substitution in bioglasses. This effective ion release can be explained considering that the strontium ions in the investigated MBG structures act as a network of modifiers and are readily bioavailable as rapid ions release to the local cell medium as it was indicated by the ICP and NMR studies.

Taking into account the in vitro results presented, we propose 2.5Sr-MBG scaffolds as the best candidate for future in vivo assays for osteoporosis diseases due to their promising result on pre-osteoblast proliferation and early markers of osteoblastic differentiation in direct contact with Sr-MBG scaffolds.

## 4. Conclusions

The three MBG powders investigated, containing 0, 2.5, and 5 mol.% of SrO, exhibited good mesopores order and high textural properties, only slightly impaired by the SrO inclusions. Moreover, MBGs exhibited in vitro bioactivity because they were coated by an apatite-like layer after 8 h (0Sr-MBG), 24 h (2.5Sr-MBG) and 72 h (5Sr-MBG) in SBF. Moreover, they underwent a progressive dissolution when soaked in MEM. Sr-containing MBGs released amounts of strontium ions able to induce early markers of osteoblastic differentiation, without reaching toxic concentrations. These features confirmed the potential of the MBGs for the production of scaffolds for bone regeneration.

A low temperature method using a PLA negative mould which was removed by extraction, allowed obtaining scaffolds based on MBG/PVA composites exhibiting interconnected giant channels, pores of around tens of microns and mesopores. Moreover, investigated scaffolds showed cytocompatibility with pre-osteoblastic cells, especially those based on 2.5Sr-MBG, in which the cytocompatibility was favored by the inductive capacity of strontium ions on proliferation and ALP gene expression of the cells. Therefore, it is concluded that 2.5Sr-MBG/PVA scaffold is an excellent candidate to be further developed to be used for bone defect treatment in non-load bearing sites.

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
