# Peer review of "Strontium-Modified Scaffolds Based on Mesoporous Bioactive Glasses/Polyvinyl Alcohol Composites for Bone Regeneration"

_materials, 2020, doi:10.3390/ma13235526_

Round 1

Reviewer 1 Report

The manuscript submitted by Jimenez-Holguin et al. describes the fabrication and characterization of strontium-modified MBG as well as MBG/PVA scaffolds for bone regeneration. The MBG are comprehensively characterized with various methods, thus, the study provides detailed information about the Sr-modified MBG which is a clear strength of the study. However, I have the following questions and points of criticism:

  1. The fabrication of Sr-modified MBG was already described in numerous other studies. Also, the combination of MBG with PVA to make it moldable/printable is not new (see e.g. Wu et al., 2011 – reference 36 in the manuscript). The authors should better explain the novelty of their work.
  2. The authors suggested the MBG/PVA scaffolds for bone regeneration e.g. in case of osteoporosis. How can I imagine are the scaffolds introduced in an osteoporotic fracture defect? Are the mechanical properties and the fast degradation of the MBG-based scaffolds as well as their printing-based pre-shaping suitable for this application?
  3. How are the MBG/PVA scaffolds stabilized after fabrication? Due to the high water-solubility, PVA is normally stabilized e.g. by thermal treatment (120-160°C). Mechanical characterization of the MBG/PVA scaffolds is missing.
  4. The discussion of the data in its current state is weak – I recommend to better explain the data with reference to each other and to the literature.

Specific comments:

  1. Page 6 / 2.4: The final dry weight of MBG in the scaffolds and/or the weight ratio of PVA vs. MBG should be provided.
  2. Page 6 / 2.4: it would be helpful for a better understanding of the macroporous structure of the MBG/PVA scaffolds, if the manufacturing process could be illustrated within a figure, e.g. by an image and/or a schematic of the PLA mould as well as the final PVA/MBG scaffold. The structure of the scaffolds becomes also not really clear from the surface images shown in Figure 7.
  3. Figure 1: maybe it would be helpful for illustration of the effect of Sr-modification, if both directions of the mesopores would be shown for all variants.
  4. Table 2 is missing. I suggest to include all tables in the main document, not in the supplementary.
  5. Figure 6 was not completely visible; statistically significant differences?
  6. For all figures showing quantitative data, the number of replicates should be mentioned in the caption.
  7. Page 16 / 383-386: the combination of MBG with PVA to obtain an extrudable biomaterial is not new – please provide reference(s) here.
  8. Page 16 / 398: intergranular macroporosity? Is it rather microporosity than macroporosity? Might be helpful to indicate such pores with an arrow in Figure 7.
  9. Figure 8A: are both images from the same sample type (2.5Sr-MBG)? It would be interesting to see a comparison between Sr-free and Sr-containing sample types. Also, an overview image from cell-seeded scaffolds would give information about the homogenitity of cell colonization.
  10. Figure 9: the „n-times“ in the y axes is rather unclear – please clarify what is meant!
  11. Page 20 / 455-458: the authors conclude that the 2.5Sr-MBG type is optimal for the cells as it provides inductive capacity of Sr ions without toxic effect. This should be explained more in detail: is the Sr concentration released from 5Sr-MBG in the toxic range? Is the Sr concentration released from 2.5Sr-MBG in the stimulatory range? The authors might refer to relevant studies in the literature. Could it be also an indirect effect of changed MBG properties due to Sr-incorporation?
  12. Page 7 / 189: sounds that the cell culture was only for 24 hours – please clarify
  13. There are a few typos/repetitions – please check the whole text again.

Author Response

Attached you will found the file containing the answers to your concerns. Thanks for your work aimed to improve the quality of the manuscript.

Reviewer 2 Report

This paper by Jiménez-Holguín and López-Hidalgo et al. is a very good example of work in the field of composite scaffolds for bone regeneration.

In my opinion this paper should be published after the following minor amendments:

1.Replace in all the document when appropriate (including the abstract) “biocompatibility” by “cytocompatibility”. Biocompatibility has a broader definition as “Biocompatibility refers to the ability of a material to perform with an appropriate host response in a specific situation”. https://doi.org/10.1016/j.biomaterials.2008.04.023. Therefore, the in vitro studies conducted in the paper in non-human cells refers to cytocompatibility.

Exception: line 73, the word biocompatibility is properly used.

2.Line 133, there are many compositions corresponding to the general term “SBF”, that are particularly critical when evaluating bioactivity. Describe the exact composition and origin of the SBF used in the study.

3.Lines 179-180, how the authors assured that the sterilization procedure was adequate. It is based in any previous study?

4. I cannot find the Table 2 data

5. Figure 5, y-axes, it is not transmittance values?

Author Response

(The authors gave the same response as above.)

Reviewer 3 Report

Jimenéz-Holuìn et. al. have developed and characterized a new mesopourous bioactive glass containing strontium at three different concentrations for applications in bone tissue engineering, particularly for bone defects.

They performed a large set of experiments to characterize the powders and scaffolds and to provide preliminary information regarding biocompatibility.

I think this investigation is of interest, and I have appreciated the number of test performed. However, I have some concerns regarding some point of the design of the study and regarding the way it is presented. Unfortunately, I believe this manuscript is not acceptable in the present form.

General

  • Minor English changes and check of the punctuation and typo errors are required: e.g. lines 26,456,500 (cellular development); 35 (including), 38 (punctuation), 44 (and so on), 73-74 (to be subjected to…), 112 (is it possible), 127 (punctuation), 142 (this), 192 (fluorescence), 208 (were), 239 (the confirmation the), 343 (as can be deduced is repeated), 457 (osteosblast), 503 (convert).
  • Why did Authors focus on osteoporosis and osteoarthritis as the major cause of large bone defects? The major cause of bone defects are others: traumatic events, tumor removal, congenital pathologies, implant revisions, or infections. Moreover, osteoarthritis causes bone lesions but not large bone defects.
  • I would suggest to uniform within the text the name of the scaffolds and ‘its components’: strontium-modified MGB/PVA obtained combining PLA negative mould and low temperature; SrO-contatining MGB, mesomacroporous quaternary SrO-containing MBG scaffolds, 0Sr-MGB, 2.5Sr-MGB, 5Sr-MGB, MGBs with composition of…, glass component. This would improve article readability.
  • A similar nomenclature has been used for both powder and scaffold: 0Sr-MGB, 2.5Sr-MGB, 5Sr-MGB. Can Authors differentiate them?
  • I would suggest to uniform within the text the name of the procedure for scaffold production: obtaining combining PLA negative mould and low temperature method for bone regeneration; low temperature method based on the MBGs-polyvinyl alcohol (PVA) system and a polymeric negative mold of polylactic acid (PLA), designed and fabricated by 3D printing; low temperature shaping method based on the system MBGs/polyvinylalcohol. This would improve article readability.

Title

I am wondering whether is it important to highlight the method of production within the title… If yes, can Authors highlight better this point through the manuscript?

Abstract

Some conclusions within the abstract section are misleading: Authors mention that “calcium, phosphorous and strontium ions are able to stimulate osteogenesis and exert a anticlastogenic action”. However, the results shown are linked to strontium release; calcium and phosphourus effects are not commented. Moreover, no experiments have been performed on monocytes or osteoclasts.

Introduction

  • Can Authors please highlight what is the innovation of their ‘formulation’ as compared to similar ones previously produced and tested? How it can have a potential therapeutic role in bone repair of bone defects?
  • Line 41: I would suggest to delete ‘in vitro’; in my personal opinion, is important to stress the biological relevance in general.
  • Line 51: I would suggest to delete ‘opposite effect’, given that inhibition of osteoclastogenesis is not an opposite effect as compared to stimulation of osteogenesis.
  • This introduction is set on the effect of strontium, while in the abstract the attention is also to calcium and phosphorous. Can Authors please discuss this point?
  • Lines 55-60: In my personal opinion the paragraph regarding the use of strontium as a bone anabolic agent for osteoporosis treatment and its use to improve bone-implant fixation is an off-topic.
  • Line 70: can Authors please discuss and provide some references to the procedure of using a 3D printed template for obtaining the desired hierarchical structure of their MBG?
  • Line 71: biocompatibility was not assessed in both disks and scaffolds.

Material and methods

  • Can Authors please provide the rationale for providing the first set of analysis only on the powders?
  • I would suggest to delete paragraph 2.3 and join it to 2.3.1, 2.3.2.
  • Can Authors please provide more details regarding the use of the transwell?
  • Can Authors please provide more details how they generate (mg/ml) data from intensities?
  • Paragraph 2.5 refers to cells cultures. Were cells cultured on PVA/MBG disks (line 179) or scaffolds (187)?
  • Can Authors please provide confocal microscopy images for 5Sr-MBG?
  • In my personal opinion, lines 200-202 are not clear enough.
  • I would suggest to join paragraphs 2.5.2, 2.5.3 and use the term cytocompatibility as used in result section 3.4.
  • LDH assays are key to provide information regarding acute toxicity: can Author please provide the same data at D3 or before this time point? Can Authors please change the term ‘colored compound (line 213), which in my opinion is inappropriate?
  • Can Authors please cite the Bragg’s law which is mentioned in result section?

Tables

Table 2 is not shown, please add it.

Results and discussion

  • Can Authors please move the sentences regarding methods in the materials and methods section, to make the text more readable?
  • Lines 243-244 are in contrast to the lines 253-255. Can Authors please rephrase the text?
  • I would suggest to increase the dimensions of all EDS spectra: the differences are hardly visible…
  • Can Authors please provide more references in the discussion (eg. lines 301, 333, 401, 418)?
  • Can Authors please discuss the loss of textural properties; as it is only briefly mentioned in conclusion (lines 484-485)?
  • Can Authors please discuss what are the implications of the delay in the formation of n-CHA?
  • In my opinion lines 367-377 are not clear enough.
  • Figure 6: is not visible the Sr release from 5Sr-MBG.
  • Can Authors please provide a discussion about the lower release of P in 5Sr-MB?
  • Can Authors please describe what do they mean for surface and fracture in figure 7?
  • In my opinion lines 428-430 are quite speculative.
  • Figure 8A: can Authors please provide larger magnifications and data on 5Sr-MBG?
  • Figure 8B: 10D is not described in the methods section; LDH was performed at D7 or D10?
  • In my personal opinion live and dead analysis are necessary to detect the amount death cells on the scaffolds.
  • Can Authors please better describe their interpretation regarding that 5Sr-MBG reduces the proliferation and do not induce ALP levels?
  • Can Authors please explain if it is known that MBG stimulates osteogenic markers expression without having been cultured in osteogenic medium? Can Authors provide some references?
  • Can Authors please better highlight the difference in the composition of the Sr-MBG present in the literature and their own?
  • In my personal opinion any reference to osteosarcoma in not pertinent.
  • In my personal opinion the use of the term ‘differentiation’ (line 467) is inappropriate, as Authors only show the expression of two markers of osteogenic differentiation. No alizarin red, von kossa assays or other functional assays have been performed.
  • I think that bioactive in vitro response in simulated body fluid should be provided also in scaffolds.
  • Can you provide come experiments and/or comments regarding the mechanical properties of these scaffolds?

Conclusion

  • 484-485, 489-490, 496-498 are not described in the discussion section; osteoclastogenesis was not analyzed
  • Can Authors please better summarize and highlight the conclusions of this study?

Author Response

(The authors gave the same response as above.)

Reviewer 4 Report

Current options for treating bone lesions present significant limitations that lead to the need for the development of innovative reconstruction synthetic grafts to support skeletal tissue repair. In this sense, present paper represents a important study with impact on this problem.

The present manuscript reported a comprehensive characterization and the discussions were logical and clear.

I suggest that the abstract be reformulated, providing readers the clearer information, e.g., the quantitative data. The topic is suitable for this journal and the data have impact on health field. The results are of interest both by fundamental research and practical approach and development of scaffolds.

Author Response

(The authors gave the same response as above.)

Reviewer 5 Report

This paper describes the potential of Strontium-modified MBG/PVA scaffold. The work could be of interest to bone tissue engineering. However, I suggest this study needs revisions before accepting for publication.

(1) According to Figure 2 C), pore size is very small. In this case, cell and tissue cannot move to inner side of scaffold. What is the advantage of this scaffold?

(2)In Figure 6, the graph of Sr of 5Sr-MBG is not shown. If possible, please add this result.

(3)In Figure 8 A), there is not the image of 0Sr and 5Sr. This image is very important result by comparing each other. Please add this information.

(4)In the case of bone regeneration, the high mechanical property is important factor. If you have data of mechanical property in this scaffold, please add new result and explain it.

Author Response

(The authors gave the same response as above.)

Round 2

Reviewer 3 Report

Dear Authors,

I have specially appreciated your effort in revising the text. However, I would suggest to enhance even more that you have designed a new way of scaffold production based on different combination of techniques and that the other major innovation consist in the addition of strontium; only few changes are required. Moreover, although I think the article have improved, I still have some concerns regarding the biological data.

  • My major concern still regards these paragraphs: (1) ‘Despite strontium is not considered an essential element, it has been demonstrated that strontium, also promotes both osteogenic proliferation and differentiation through several pathways such as Wnt/ β-Catening signaling [23], Ras/MAPK signalling [24] and the Ca sensing receptor (CasR) [25]. On the other hand, it is well known that strontium has the ability to inhibit osteoclastogenesis, blocking the interaction between Rank ligand and receptor [26], inhibiting the osteoclasts differentiation [27] and promoting their apoptosis mediated by CasR [28]. In this sense, this element was introduced into clinical practice in the form of oral strontium ranelate, a widely used prescription against osteoporosis [29]. Oral administration of strontium improves bone-implant fixation. However, a constant "in situ" release of this ion directly into the implant-tissue interface is more effective, as it simultaneously stimulates bone formation while minimizing side effects potentially related to high oral doses of strontium [30,31].’; (2) ‘Taking into account these in vitro results, we propose 2.5Sr-MBG scaffolds as the best candidate for future in vivo assays for osteoporosis or osteoarthritis diseases due to their promising result on pre-osteoblast proliferation and differentiation in direct contact with Sr-MBG scaffolds’.

I can understand that Authors have focused on osteoporosis and osteoarthritis due to the Sr therapeutic advantages. However, I do believe that this point should emerge in both introduction and discussion section and any possible misunderstandings should be avoided. Currently, the introduction is calibrated on bone defects.

I’m aware of the effects of strontium ranelate on osteoblast and osteoclasts and why it has been proposed for the therapy of osteoporosis. I’m also convinced can this molecule can improve bone implant fixation or stimulate bone regeneration in bone defects, especially in a context of osteoporotic bone. Furthermore, I can agree that a local administration of strontium could be preferable to a systemic administration when the objective is to enhance the osseointegration of implants-scaffolds. However, how the paragraph is written (and according to what Authors has answered to my comments on lanes 55-60), it could seem that a local release of strontium can be more effective as compared to a systemic administration when generally prescribed against osteoporosis. In my opinion, a local release from a scaffold should not be considered as a therapeutic alternative to systemic administration of a drug in such a systemic pathology. What I was trying to communicate is that any references to osteoporosis should be limited to the specific issue of bone regeneration within an osteoporotic context. Any other reference could create misunderstandings.

Moreover, why these scaffolds should be considered for osteoarthritis has been answered by Authors only with a reference. If this point is not contextualized and discussed, I think that could only create confusion. Any reference to osteoarthritis should be limited to the osteochondral defect reconstruction as the biological data shown here refers only to ‘osteogenesis’. Strontium has been effectively tested in preclinical studies for the therapy of osteoarthritis, thus this scaffold could be proposed as a scaffold delivering a factor counteracting osteoarthritis. However, these effects are related to modification of cartilage, modulation of pain and anti-inflammatory effect as well as modulation of subchondral bone remodelling, does not relate to the treatment of bone defects which is the topic of the present article. This point is the one which could create some misunderstandings.

  • Currently, the article is proposing a new scaffold formulation for bone regeneration in bone defects, particularly for osteoporosis. First, several osteoporotic fractures do not require at all the use of bone substitutes. Second, the vast majority of osteoporotic fractures are fragility fractures due to load-bearing and compression. In this context, I do believe that within the text should emerge the limitation of mechanical properties of this scaffold and that the application should be considered in non-load bearing sites.

  • I totally disagree with Authors’ response regarding LDH assays. Lactate dehydrogenase is an enzyme found in nearly all living cells.  In response to cellular damage, that can be induced in this specific case by toxic components of the scaffolds, LDH is released from the cytoplasm into the extracellular environment and detected in the medium by LDH assays. I didn’t get what do Authors mean as ‘the expression of LDH usually occurs late and can be better appreciated with enough reliability over long periods such as a week’. The analysis at short time points is widely diffused. Moreover, usually in these type of experiments a positive control of toxicity is used to confirm the reliability of the assay. By performing LDH assays only at D7, Authors assayed a long-term toxicity and did not investigated the acute toxicity. We do not have any confocal data at D7 of 5Sr-MBG scaffolds, therefore we cannot exclude that the majority of cells can be death early and thus do not release high levels of LDH at D7. Furthermore, I do think that the hypothesis formulated (‘Can be explained taking into account that the previously observed high proliferation led to a situation of early confluence, lack of space and, therefore, the increment of cell death and the value obtained during LDH measurement’) is not congruent with that LDH levels is similar in 0Sr-MBG scaffolds, 2.5Sr-MBG scaffolds, 5Sr-MBG scaffolds (the difference is not statistically significant). Moreover, did not the cells colonize the inner parts of the scaffold? Why do Authors think that confluence should necessarily induce cells death? Cells stops to proliferate also when they are differentiating…

  • I do not believe that the conclusion ‘The examination of the preosteoblasts actin and DAPI reveals that pre-osteoblasts presented their typical spindle-shaped morphology and were adequately spread on the scaffold surface, indicating their cytocompatibility.’ can be drawn based only on one image of two cells of one scaffold.

  • Authors affirm to have changed differentiation term throughout the text; however, I still found several references to differentiation and osteoblastogenesis eg. 27, 723

Few other suggestions:

  • In the introduction the fact that SiO2-P2O5-CaO based scaffolds has osteoinductive effect produced by P and Ca release can be specified (line 50)
  • The increased release of calcium ions in Sr-MBG scaffold is not ‘slightly’ (line 547)
  • Figure 6: the comparison between 2.5Sr-MBG and 5Sr-MBG is not clear and seems not coherent with the figure.
  • Placing MBG disk on a plate with transwell to obtain aliquots of medium without MBG grains can be specified (line 218)
  • Line 74: by blocking the interaction between Rank ligand and its receptor, and promote osteoclast apoptosis mediated by CasR.
  • 3d printing > a 3d printed mould (line 577)

Moderate English changes are required, I still found several errors:

  • Resulting of > resulting from (line 15)
  • Revision of infection > revision for infection (line 17)
  • Its > their (lines 66; 148; 576)
  • Are > is (line 91); Their > its (line 92)
  • Will > were (line 106); Will > was (line 112); Allows > would allow (line 117); Produce > would produce (line 119)
  • Replicas > replicates (line 212, 224, 277)
  • Lines 245-247; 674-677: unclear
  • I would suggest to uniform the tenses of results and discussion section to the other sections: Involves > involved (line 257); Is > was (line 259); Were > was (line 288), were > was (line 300); Allow > allowed (line 332); confirms > confirmed (line 367); decrease > decreased (line 392); Increase > increased; induce > induced; noticeable decreases > a noticeable decrease (line 393); confirms > confirmed (line 456); become > became (line 458); confirm > confirmed (line 594)
  • DISCUSION > DISCUSSION (331)
  • That in > that also in
  • Exhisrimentally > experimentally (line 347)
  • He > the (line 364)
  • Dp is repeated (line 395)
  • Reported > reports (line 396)
  • Get > found (line 405)
  • Anther a > another (line 488)
  • Thickens > thickness (line 512)
  • Ionss > ions (line 518)
  • Distribution: > distribution (macropores and giant channels) (lines 592-593)
  • Discs > disks (line 654)
  • Delete ‘as it was demonstrated’ (line 701)
  • Cytocompatiblity > cytocompatibility (line 735)

Author Response

Attached you will found the file containing the answers to your concerns. Thanks for your work aimed to improve the quality of the manuscript

Reviewer 5 Report

This paper describes the potential of Strontium-modified MBG/PVA scaffold. The work could be of interest to bone tissue engineering.

Author Response

The attached version of the manuscript (R2 version) was greatly improved compared R1 version after taking into consideration the comments and concerns of the reviewers Please find highligted in yellow all the modifications regarding the R1 version,

 Thanks for your work aimed to improve the quality of the manuscript